# Factors Influencing Abundances and Population Size Structure of the Threatened and Endemic Cyprinodont *Aphanius iberus* in Mediterranean Brackish Ponds

**Serena Sgarzi** [1,*] , **Sandra Brucet** [1,2,*], **Mireia Bartrons** [1] , **Ignasi Arranz** [1,3], **Lluís Benejam** [1] and **Anna Badosa** [1]

1   Aquatic Ecology Group, University of Vic—Central University of Catalonia, 08500 Barcelona, Spain; mireia.bartrons@uvic.cat (M.B.); ignasi.arranz-urgell@univ-tlse3.fr (I.A.); lluis.benejam@uvic.cat (L.B.); anna.badosa@uvic.cat (A.B.)
2   Catalan Institution for Research and Advanced Studies (ICREA), 08010 Barcelona, Spain
3   Laboratoire Evolution et Diversité Biologique (EDB UMR 5174), Université de Tolouse, CNRS, IRD, UPS, 118 Route de Narbonne, F-31062 Tolouse, France
*   Correspondence: serena.sgarzi@uvic.cat (S.S.); sandra.brucet@uvic.cat (S.B.); Tel.: +34-93-881-5519 (S.S. & S.B.)

**Abstract:** *Aphanius iberus* is an endemic cyprinodontoid fish species of Mediterranean ponds in danger of extinction. In this study, we studied some abiotic and biotic factors that can influence *A. iberus*'s size structure and density in Mediterranean brackish ponds. We sampled fish using fyke nets in 10 ponds of Empordà (Spain) during the spring season. Our results showed that a better ecological status (according to the Water Quality of Lentic and Shallow Ecosystems (QAELS) index), pond's depth and pond's isolation (reflected by an increase in total nitrogen) were related to larger individual sizes and more size-diverse populations. Increasing the salinity is known to help the euryhaline *A. iberus* acting as a refuge from competitors. Nevertheless, our results showed that higher conductivities had a negative effect on *A. iberus*'s size structure, leading to a decrease in the mean and maximum size of the fish. Fish abundance (expressed as captures per unit of effort (CPUE)) seemed to increase with increasing the pond's depth and total nitrogen (the latter reflecting pond isolation). In conclusion, our results suggest that achieving a better pond ecological status may be important for the conservation of endangered *A. iberus*, because better size-structured populations (i.e., larger mean and average lengths) were found at higher water quality conditions.

**Keywords:** Mediterranean ponds; fish; *Aphanius iberus*; size structure; ecological status

## 1. Introduction

Aquatic ecosystems of the Iberian Peninsula are a hotspot for endemic freshwater fish fauna; still, most of the fish species are critically threatened by habitat destruction, intensive agricultural activities or the introduction of exotic species [1,2]. The Spanish toothcarp (*Aphanius iberus*, Valenciennes, 1846) is a small cyprinodont (up to 6 cm in length) endemic from the Eastern Mediterranean lowland waters of the Iberian Peninsula [3–5] and in danger of extinction [6,7]. As other cyprinodonts, it is characterized by fast growth, early maturity, high reproductive effort and multiple spawnings [8,9], which implies a short longevity (age up to two+). *A. iberus* is an eurytherm and euryhaline species, well-adapted to changes in environmental conditions [10] such as sudden alterations in temperature and salinity due to marine intrusions or freshwater floodings [11]. This cyprinodont originally inhabited a wide range

of lowland waterbodies, but now, its geographical distribution is limited to brackish and hypersaline coastal waterbodies [1,5,8,12] due to habitat degradation (e.g., intensive agriculture, water pollution and wetland desiccation) and the introduction of invasive species, which usually act in an additive manner, since habitat degradation facilitates biological invasions [13,14]. The high degree of isolation among the remaining populations also poses a threat to their conservation, as they show higher rates of extinction than populations in well-connected locations [5].

Some studies have shown that the abundance and size structure of *A. iberus* depends on the ecological status of the ponds, with larger individuals and higher densities found in ponds with a higher water quality [15]. Indeed, in Italy, another species of the *Aphanius* genus (*Aphanius fasciatus*) has been proposed as an indicator of the ecological status of salt marshes [16], suggesting that those fish are sensitive to changes in the ecological status of their environments. More confined and less accessible ponds also seem to host populations of *A. iberus* more abundant and stable over time [17]. Another variable that may influence the density and size structure of *A. iberus* is the pond morphometry (area and depth), because it has a strong impact on the structural complexity and niche availability, as has been found for other fish species and communities [18–20]. However, to our knowledge, there have been not studies in this respect.

The Eastern mosquitofish (*Gambusia holbrooki*) [21,22] is an invasive species that very often interacts with *A. iberus*, because both share similar habitats [23] and compete for the same resources [24,25]. Both fish species are zooplanktivorous, but *G. holbrooki* consumes mainly cladocerans, ostracods and copepods [26,27], and *A. iberus* prefers harpacticoid copepods, copepod nauplii and detritus [24]. Sometimes, *G. holbrooki* can act in an aggressive way against *A. iberus,* and this behavior seems to be inversely proportional to the salinity, as well as its ability to capture prey [28]. Young individuals of *A. iberus* have been found to capture less prey in the presence of conspecific adults and *G. holbrooki*, suggesting both strong intraspecific and interspecific competition [12,29]. Currently, *A. iberus* has disappeared from fresh and oligohaline waters, and its habitat is restricted to salt marshes, coastal lagoons and river mouths [24,30], where the invasion success of the mosquitofish is limited due to the high salinity fluctuations [24,31,32]. Nevertheless, although *A. iberus* tolerates high salinity conditions, its metabolism may be affected when the salinity levels in the location are high [33–35]. Physiological functions such as oxygen consumption, critical swimming speed and routine activity level show a general decrease at the extreme salinity in *Aphanius dispar* [36], although the spawning efficiency seems not to be significantly affected by the changes in salinity [37].

Identifying the key factors that influence the population structure of *A. iberus* is relevant to develop efficient conservation and management plans for this endangered species. The body size of *A. iberus* has been used to assess growth-related parameters, such as age [8], fecundity and sexual maturity [38], as well as ammonia excretion rates [39]. Although, in Mediterranean brackish ponds, trophic interactions are very often body size-dependent [40–42], studies about the size structure of *A. iberus* and the factors that determine it are scarce.

The present study aims to identify the factors influencing the abundances and population size structure of A. *iberus* in the north-east of the Iberian Peninsula in late-spring (i.e., when this species finishes the first period of annual reproduction). Specifically, we assessed whether abiotic factors (i.e., conductivity, nutrient concentrations and pond morphology); the ponds' ecological status; food resource availability (zooplankton biomass) and the presence of the main competitor, *Gambusia holbrooki*, are correlated with the size structure and abundance of this endangered species in 10 coastal brackish and hypersaline ponds. We assessed the size structure using the size diversity index [43], in addition to several size metrics, such as the maximum size, mean size and size range.

We hypothesized that a good ecological status, together with a larger pond dimension (depth and area), would increase the possibilities to find well size-structured populations with a higher size diversity, as well as higher densities of fish, as these two factors are supposed to set good conditions for the fish growth. Concerning conductivity, we expected a decrease in the size-related variables of *A. iberus*, because the high conductivity negatively affects its metabolism. In contrast, locations at

higher conductivity levels could host higher *A. iberus* densities, because high conductivity may prevent the colonization of invasive species, such as *G. holbrooki*. We also hypothesized that the presence of *G. holbrooki* (main competitor of *A. iberus*) would lead to lower densities of *A. iberus*, as *G. holbrooki* have been observed to outcompete *A. iberus* [30]. Finally, we would expect that the abundance of *A. iberus* would be negatively correlated with the zooplankton biomass due to fish predation on zooplankton.

## 2. Materials and Methods

### 2.1. Study Area

The studied ponds are located in two protected areas of the Empordà coastal wetlands (Figure 1), between 42°01′42″ N–3°11′18″ E and 42°15′58″ N–3°08′17″ E of the Ter River Basin (NE Iberian Peninsula). Eight of the ten ponds were located in the "Aiguamolls de l'Empordà" Natural Park, and two were located south in the "El Montgrí, Illes Medes i el Baix Ter" Natural Park (some examples of individual ponds are shown in Figure S1. The climate is Mediterranean, with hot, dry summers and mild, wet winters. The hydrology of these Mediterranean coastal wetlands is characterized by a prolonged confinement period during warm seasons without water inputs, followed by irregular flooding events (i.e., rainfall or marine intrusions during sea storms, the latter no more often than twice a year) [44]. Hydrological connections among ponds and/or to rivers and the sea take place only during such irregular flooding events [45]. The studied ponds are characterized by their shallowness and cover a range of morphometry (e.g., pond area and depth) and conductivity [46]. Concentrations of inorganic nutrients (nitrates and phosphates) in late-spring are low due to the scarceness of water inputs, but concentrations of total nutrients, especially total nitrogen, are high due to a concentration effect produced by a high evaporation rate [44,45,47]. In these confined coastal environments, nitrogen, rather than phosphorous, usually limits the primary production [41,48,49]. For more details about the environmental characteristics and planktonic composition of the studied ponds, see [46].

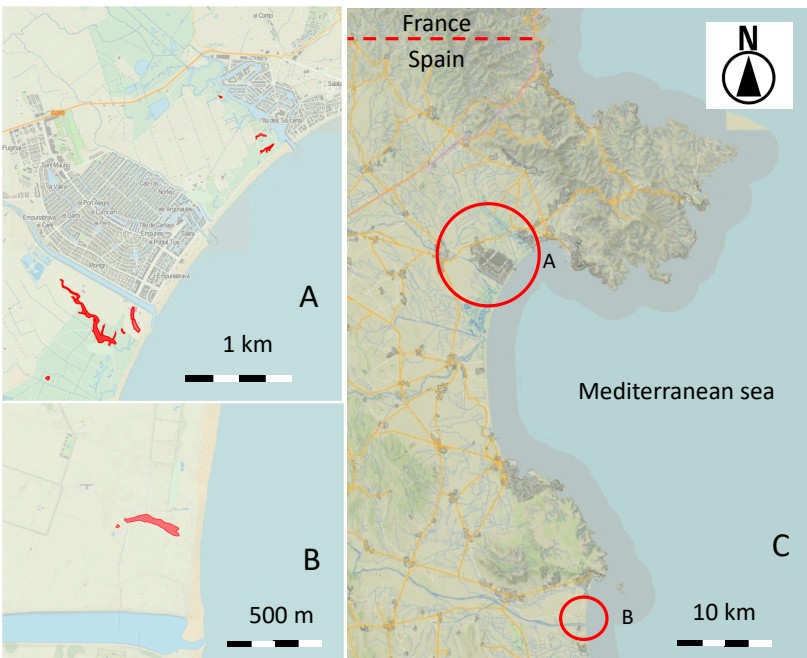

**Figure 1.** Location of the study region: Empordà coastal wetlands (Ter River Basin in the NE of the Iberian Peninsula) (**C**), with the detailed geographical position on the studied ponds in red (**A**,**B**). Eight of the studied ponds were located in the "Aiguamolls de l'Empordà" Natural Park (**A**), and two were located south in the "El Montgrí, Illes Medes i el Baix Ter" Natural Park (**B**). This map was produced with the online software ArcGIS (version 10.5.1, 2017, ESRI Environmental System Research Institute, Redlands, CA, USA) (https://www.arcgis.com).

Fish community in the studied ponds is mainly composed by *Aphanius iberus* and the invasive fish species *Gambusia holbrooki* [46]. Whereas the former is, overall, more abundant in ponds of higher salinity, the latter is more abundant in oligohaline ponds [46]. The rest of the community is composed by the *Atherina boyeri* (Risso, 1810), *Pomatoschistus* sp. (Gill, 1863), *Mugil cephalus* (Linnaeus, 1758), the marine fish *Solea solea* (Queusel, 1806), *Anguila anguila* (Linnaeus, 1758) and the invasive *Lepomis gibbosus* (Linnaeus, 1758), although the last three species are very scarce (see [46] for more details on the relative abundance of the fish species).

## 2.2. Field Sampling and Analysis

The 10 ponds studied were sampled once during the end of the spring season (from May to early June 2016). Conductivity (mS·cm$^{-1}$) was measured using a multiparameter probe (Hanna Instruments, Woonsocket, RI, USA). Total area (m$^2$) of each pond was estimated by using the "Google Maps Area Calculator Tool" [50] (reference 58, old paper), while the mean water column depth (cm) was calculated from in situ repeated measures obtained with a two-meter rule. Total nitrogen (mg·L$^{-1}$) was measured according to Koroleff, 1973 [51], adapted by Seal Analytical to an integrated system of a CFA (Continuous Flow Analysis) digester.

Two ecological indices related to the pond's ecological status were used in each pond: (1) the ECELS (Conservation Status of Lentic and Shallow Ecosystems) estimates the conservation status of lentic ecosystems based on morphological aspects, type of aquatic vegetation and human impacts [52]. The ECELS categories range from bad (0–30 out of 100), deficient (30–50 out of 100), mediocre (50–70 out of 100), good (70–90 out of 100) and very good (90–100 out of 100), and (2) the QAELS (Water Quality of Lentic and Shallow Ecosystems) index evaluates the water quality based on the composition of microcrustacean assemblages and taxonomic richness of aquatic insects and crustaceans in the Mediterranean wetlands [53]. The QAELS categories range from bad (<0.46), deficient (0.46–0.55), mediocre (0.55–0.62), good (0.62–0.72) and very good (≥0.72). The QAELS index was calculated after the observation of macroinvertebrate samples under optic microscope and a stereoscope. Samples were obtained through a dip net (mesh size 250 μm) following standard protocols [53].

Zooplankton samples were taken from each pond by mixing subsamples from five different sites in order to overcome the expected patchy distribution of plankton. Five liters of mixed water samples were filtered through a 50-μm mesh size net and preserved in 4% Lugol's acid solution. Zooplankton individuals collected (including rotifers, copepods and cladocerans) were counted, identified and measured using a stereoscope and an inverted microscope (Utermöhl method), as was described in [46]. To estimate the zooplankton biomass, the total length (μm) of the first 100 individuals (when possible) was measured assuming that all individuals were equally distributed in the observed sample. Individual biomasses were then calculated using approximation to shape formulas.

Fish were caught by fyke nets set for 24 h, a common and widely used method in coastal lagoons [17,54,55]. Fyke nets consisted of a semicircular entrance ring followed by three smaller circular rings surrounded by a net (3.5-mm mesh) with two consecutive funnels (120 mm of funnel diameter, 1050 cm$^2$ of interception area, 98 cm of length, 30 cm of height and 95 cm of wing length). The total number of fyke nets set in each pond varied according to its area and depth (Table S1). A total of 49 fyke nets were set in all the ponds. All captured fish were sexed (except juveniles < 13 mm), measured for total length (mm) and released. We measured all the individuals of *Aphanius iberus* in each sample, in order to minimize the error estimation.

## 2.3. Aphanius iberus Abundance and Size Structure

In each pond, the *Aphanius iberus* abundance was calculated by dividing the total captures by the number of fyke nets set in each pond (captures per unit of effort; CPUE). The size structure of *A. iberus* in each fyke net was assessed using four size-based metrics: (1) the maximum size, (2) mean size (computed as the geometric mean), (3) size range (the difference between the maximum and the minimum size) and (4) size diversity index (μ). For each fyke net, the size diversity was calculated

using individual size (i.e., length) measurements, as proposed by [43]. Size diversity is based on the Shannon–Wiener diversity index [56] adapted for a continuous variable, such as body size. This index is the continuous analog of the taxonomic Shannon diversity index, and it produces values in a similar range to those of the Shannon index. In our case, it integrates the amplitude of the length range and relative abundance of the different lengths. Thus, the high values of the size diversity would indicate a high diversity of sizes with an equitable numerical frequency of sizes along the distributions [57,58]. In contrast, the low values of size diversity (rarely taking negative values) would indicate a low diversity of fish sizes with an inequitable numerical frequency of sizes along the distribution [43].

### 2.4. Data Analysis

We used mixed linear models (MLMs) to test the effects of abiotic and biotic factors on the abundance and size metrics of *A. iberus*. We considered captures of each fyke net as an observation unit (N = 49), and "pond" was introduced as a random effect to deal with pseudoreplication. As predictor variables, we considered conductivity, pond area and mean water depth, total nitrogen, ECELS and QAELS indexes (as estimates of ecological status) and zooplankton biomass (as food resource availability). Pearson's r index revealed correlations among some of those variables. We applied Bonferroni correction to counteract the multiple comparisons issue and, finally, removed the variables that were highly correlated (>0.6). As *G. holbrooki* was not present in all the ponds (it was absent in 7 ponds out of 10), we could not include its abundance in our MLMs. Instead, we performed an ANOVA, comparing the situations "with" and "without" the presence of *G. holbrooki* for the abundance and for each of the size-related variables of *A. iberus* considered in this study.

We looked for the most parsimonious model from the full models by using a stepwise (backward) selection. The most parsimonious model was chosen using the Akaike information criteria (lowest AIC), which represents the best at explaining the data with the lowest combination of variables. We also calculated the standardized (beta) coefficients for the significant predictors included in the best models by using the R package "QuantPsyc" version 1.5 [59]. Predictors were previously checked for normality and homogeneity of variance, and, if variables did not meet the assumptions, base 10 logarithmic transformations were applied. Additionally, a visual inspection of the residual plots was done to detect any violation of the regression assumptions. In order to improve homoscedasticity, we used the function "varPower" of the package "nlme" [60]. For the creation of the boxplots, we used the "ggplot2" package [61]. All analyses were done with the software R version 3.4.2 (R core Team, 2017, Boston, MA, USA).

## 3. Results

### 3.1. Description of the Local Characteristics in the Mediterranean Ponds

There were wide ranges of environmental and ecological conditions across the study area (Table 1). Abiotic and biotic factors measured, as well as ECELS and QAELS indexes of ecological status, are shown in Table 1. The 10 studied ponds during spring showed conductivity values ranging from 10.7 mS·cm$^{-1}$ to 69.10 mS·cm$^{-1}$ and mean water column depths ranging from 16 cm up to 150 cm. The ponds differed quite a lot in their areas, with values ranging from 147.90 m$^2$ to 68,150 m$^2$. Total nitrogen, which includes organic and inorganic nitrogen compounds, showed the lowest value at 55.58 μmol·L$^{-1}$ and the highest value at 234.40 μmol·L$^{-1}$. The zooplankton biomass ranged from 1.13 μg·L$^{-1}$ up to 4840.48 μg·L$^{-1}$. Regarding the ponds' ecological status, ECELS index values ranged from 43 to 98, indicating a "bad" and "mediocre" status, respectively. Whereas the QAELS index ranged from 0.25 to 0.56, also indicating a "bad" and a "mediocre" status, respectively (Table S1). As an average, ponds showed a "deficient" status according to the mean index value.

**Table 1.** Mean, standard deviation (SD), minimum and maximum values of the abiotic and biotic factors and ecological status indexes measured in the study ponds (N = 10). ECELS: Conservation Status of Lentic and Shallow Ecosystems and QAELS: Water Quality of Lentic and Shallow Ecosystems.

|  | Mean | SD | Minimum | Maximum |
|---|---|---|---|---|
| Mean water column depth (cm) | 59.30 | 41.93 | 16.00 | 150.00 |
| Pond area (m$^2$) | 10,873.60 | 20,463.50 | 147.90 | 68,150.00 |
| Conductivity (mS·cm$^{-1}$) | 46.14 | 17.63 | 10.07 | 69.10 |
| Total nitrogen (μmol·L$^{-1}$) | 92.01 | 51.63 | 58.55 | 234.40 |
| Zooplakton biomass (μg·L$^{-1}$) | 498.17 | 1313.92 | 1.13 | 4840.48 |
| ECELS index | 77.00 | 18.70 | 43.00 | 98.00 |
| QAELS$^e$ 2010 index | 0.47 | 0.12 | 0.25 | 0.56 |

*3.2. Variation of the Population Structure of A. iberus across Mediterranean Ponds*

The mean, standard deviation and minimum and maximum values calculated for the different size metrics of *A. iberus*, as well as the abundance (expressed as CPUE), are shown in Table 2. Maximum length values ranged from 19 mm to 54 mm, while the mean length had a minimum of 16 mm and a maximum of 41 mm. This last value coincided with the maximum value of the length range, whereas the minimum length range value was 6 mm. Size diversity showed a wide range of values, from 0.27 to 2.34. *A. iberus* abundance (in CPUE) also varied largely among fyke nets, from two individuals to 525 individuals.

**Table 2.** *A. iberus* size metrics and abundance (CPUE) obtained/computed per each sample (N = 49). The descriptive statistics are the mean, standard deviation (SD), minimum and maximum.

|  | Mean | SD | Minimum | Maximum |
|---|---|---|---|---|
| *Aphanius iberus* maximum length (mm) | 41.08 | 8.06 | 19.00 | 54.00 |
| *Aphanius iberus* mean length (mm) | 29.21 | 5.36 | 16.00 | 41.00 |
| *Aphanius iberus* length range (mm) | 21.67 | 7.38 | 6.00 | 41.00 |
| *Aphanius iberus* size diversity | 1.15 | 0.36 | 0.27 | 2.34 |
| *Aphanius iberus* capture per effort unit (CPUE) | 59.49 | 91.68 | 2.00 | 525.00 |

*3.3. Main Drivers Affecting the Fish Population Size Structure and Density*

The MLMs identified the most important drivers influencing the fish population size structure and density across all ponds. The most parsimonious significant models for each fish metric mentioned above (Table 2) as dependent variables are shown in Table 3. The results showed that the maximum length was negatively related to the conductivity and zooplankton biomass but positively related to the total nitrogen and QAELS index. The zooplankton biomass was the predictor with the strongest effect on the maximum length of *A. iberus* (Table 3).

Concerning the mean length of the fish, the MLM model showed similar results as when considering the maximum length of *A. iberus* as a response variable. The mean length of *A. iberus* significantly decreased with the increasing conductivity and zooplankton biomass but showed a positive relation with the ecological quality index QAELS.

**Table 3.** Results of the linear mixed models (N = 49 fyke nets) showing the predictor variables that significantly relate with *Aphanius iberus* size metrics and abundance (expressed as CPUE). Only the most parsimonious significant models were shown for each response variable. For each model, the intercept (estimated and standard error, S.E.), beta coefficients (standardized), *t*-value, significance (*p*-value) and degrees of freedom (df) are also reported.

| Response Variable | Predictor | Estimate | S.E. | Beta Coefficients | *t*-Value | *p*-Value | df |
|---|---|---|---|---|---|---|---|
| *Aphanius iberus* MAXIMUM length | Conductivity | −19.49 | 4.69 | −0.47 | −4.15 | <0.01 | 5 |
| | Log Total Nitrogen | 2325.09 | 564.20 | 0.45 | 4.12 | <0.01 | 5 |
| | QAELS index | 3894.26 | 690.27 | 0.59 | 5.64 | <0.01 | 5 |
| | Zooplankton biomass | −0.22 | 0.05 | −0.69 | −4.51 | <0.01 | 5 |
| *Aphanius iberus* MEAN length | Conductivity | −0.15 | 0.05 | −0.42 | −3.35 | 0.02 | 6 |
| | QAELS index | 31.94 | 6.76 | 0.55 | 4.73 | <0.01 | 6 |
| | Zooplankton biomass | −0.01 | −0.01 | −0.59 | −4.01 | 0.01 | 6 |
| *Aphanius iberus* length RANGE | Log Pond Mean Depth | 14.45 | 5.36 | 0.42 | 2.70 | 0.03 | 7 |
| | Log Total Nitrogen | 36.88 | 8.93 | 0.48 | 4.13 | <0.01 | 7 |
| *Aphanius iberus* SIZE DIVERSITY | Conductivity | <0.01 | <0.01 | 0.29 | 2.21 | 0.06 | 7 |
| | Log Total Nitrogen | 0.93 | 0.37 | 0.30 | 2.51 | 0.04 | 7 |
| *Aphanius iberus* capture per effort unit (CPUE) | Log Pond Mean Depth | 1.46 | 0.57 | 0.55 | 2.56 | 0.04 | 7 |
| | Log Total Nitrogen | 2.15 | 0.97 | 0.36 | 2.23 | 0.06 | 7 |

The length range of *A. iberus* was found to be positively related to the pond mean depth and total nitrogen. Beta coefficients of this model (0.42 and 0.48) showed similar effects of both predictor variables on the length range. The same results were found for fish density, with the CPUE positively related to the mean depth and total nitrogen (Table 3), suggesting that a higher number of individuals inhabited more nutrient-rich and larger ponds. However, the pond mean depth exhibited a stronger correlation on the fish density than the total nitrogen (beta coefficients of 0.55 and 0.36, respectively; Table 3).

Concerning the size diversity, it was positively related to the total nitrogen and only slightly related with the conductivity (Table 3). In this case, the beta coefficients for the two main drivers were similar (0.29 for conductivity and 0.30 for total nitrogen; Table 3). Finally, the ECELS index and pond area were the only variables not retained in any of the models selected (see the full models in Supplementary Table S2).

### 3.4. Influence of the Presence of G. holbrooki on the Size Structure and Density of A. iberus

Boxplots showed the CPUE and size metrics of *A. iberus* in the presence and absence of *G. holbrooki* in the pond (Figure 2). Overall, the presence of *G. holbrooki* in the pond did not significantly modify the size structure and density of *A. iberus* (*p*-values > 0.39; Figure 2). However, the mean size of *A. iberus* was significantly higher when *G. holbrooki* was present in the pond (Figure 2), indicating an unexpected increase of body size with the presence of the main competitor.

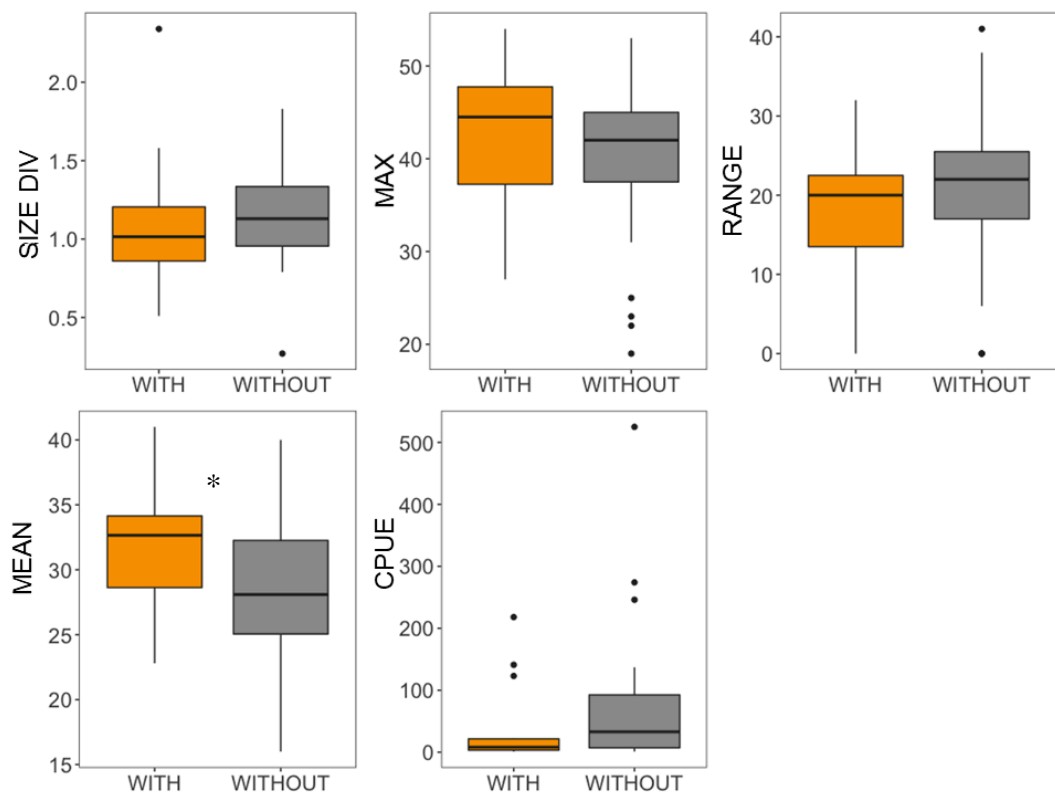

**Figure 2.** Boxplots showing the distribution of *Aphanius iberus* capture per unit of effort (CPUE) and size-related variables according to the presence ("with Gambusia") or absence ("without Gambusia") of *Gambusia holbrooki*. Significant differences are marked with the asterisk symbol (*).

## 4. Discussion

Our results suggested that both the maximum and mean sizes of *A. iberus* increased with the increasing pond's water quality (QAELS index) and decreased with the increasing conductivity and zooplankton biomass. The size range, maximum size, size diversity and CPUE of *A. iberus* were positively related to the nutrient concentration (i.e., total nitrogen), while the size range and CPUE were also larger in deeper ponds. In contrast to our hypothesis, the presence of *G. holbrooki* seemed not to affect negatively the population structure of *A. iberus*.

We found larger maximum and mean lengths of *A. iberus* in locations with better water quality (i.e., the QAELS index). These results support a previous study on multiple water bodies in the southernmost distribution area of *A. iberus*, showing its preference for ponds of better ecological status [15]. Although the results showed that the average of the studied locations had a deficient ecological status, higher values of the QAELS index are usually associated with the predominance of large zooplankton (such as big copepods) over small rotifers (that are more linked to eutrophic and hypoxic conditions; [53,54,62]. Adults of *A. iberus* (with larger body lengths) are usually associated with glasswort habitats (highly productive and occasionally inundated environments) where big zooplankton is more abundant, and this may support the positive relationship between *A. iberus* size and the ecological status of the ponds found in our study. In contrast, younger and smaller individuals positively select more eutrophic algal mats, associated with a bad ecological status, where small rotifers dominate [24].

The total nitrogen was found to be related with the CPUE, as well as with all size metrics, expect the mean length. In Mediterranean salt marshes, the concentration of total nitrogen in the water is an indicator of a pond's confinement or isolation [44], and during the late-spring and summer season, the total nitrogen is more concentrated because of evaporation processes [47]. Our results are in accordance with previous studies that showed that, in more confined and less accessible ponds,

the populations of *A. iberus* are more abundant and stable over time, probably due to a lack of external perturbations, such as isolation from invasive species [17] or fewer entries of freshwater inputs. Since nitrogen is typically the limiting nutrient in these ponds, a higher total nitrogen concentration could also be associated with higher production rates of *A. iberus*, which, in turn, may favor larger populations.

Our results also showed, as expected, a negative relationship among conductivity and the maximum and mean sizes of *A. iberus*. In high saline habitats, conductivity can act as a "refuge" for the *A. iberus* to avoid the colonization of less salt-tolerant fish species, such as *G. holbrooki* [63]. However, high salinity levels may also have negative effects on the metabolism of cyprinodontoids [5,36], because the energy used for osmoregulation is not available for their growth performance and survival [33–35]. This could explain the presence of smaller fish in ponds at higher levels of conductivity.

In our study, the *A. iberus* CPUE, along with the *A. iberus* size range, were positively related to the pond depth. Similarly, studies from European wetlands and lakes found that wider and deeper waterbodies hosted greater biomass and sizes of fish [17,64], with the consequent higher probability to find a wider range of fish sizes [20]. Additionally, individuals of *A. iberus* trapped in brackish ponds due to competition exclusion and habitat degradation can, in some cases, reach unnaturally high densities [17].

Our results also showed that the zooplankton biomass is negatively correlated to the *A. iberus* maximum and mean sizes. Other studies on *A. iberus* observed that both juveniles and adults of this species have similar food preferences, as they mainly feed on harpacticoids, copepods and nauplii, detritus and diptera larvae [24]. Still, smaller individuals prefer feeding on small-sized prey, while larger fish show a greater preference for large-sized prey [24]. Larger individuals have higher feeding rates [34,65]. Thus, the presence of larger fish (expressed by higher mean and maximum sizes) may imply a lower zooplankton biomass, as it increases the consumption rates with the fish body sizes. In addition, previous studies observed that, when the potential resource availability is low, the fish size distribution tends to be more diverse, suggesting that competitive interactions for resources promote diversification by size [19,66,67].

As for the influence of *G. holbrooki* on *A. iberus* abundance and size structure, the results suggested that both the CPUE and size metrics of *A. iberus* were not affected by the presence of this allochthone fish. Only the mean size seemed to be slightly positively affected by the presence of the competitor. This result differed from our expectations, in which *A. iberus* would be smaller and less abundant in the presence of *G. holbrooki*. This apparent inconsistency could be explained by the fact that *G. holbrooki* was found just in few of the studied ponds, the ones with lower conductivity, and what we observed could be an indirect effect of environmental conditions that favor *A. iberus* development more than the effect of direct competition. Thus, the low number of ponds with *G. holbrooki* in our study did not unable us to derive strong conclusions about the influence of the *G. holbrooki* presence on *A. iberus* abundance and size structure.

In conclusion, our results suggest that the ponds' ecological status (as shown by the QAELS index), depth, conductivity and nutrient concentrations are key variables that determine the variations of the size structure and abundance of *A. iberus* in Mediterranean brackish ponds. Achieving a better pond ecological status seems to be important for the conservation of endangered *A. iberus,* because better size-structured populations (i.e., larger mean and average lengths) are found at higher water quality conditions. In addition, a pond's isolation may also be an advantage to preserve *A. iberus* populations.

**Supplementary Materials:** The following are available online at http://www.mdpi.com/2073-4441/12/11/3264/s1: Figure S1: title, Table S1: Main geographic and morphometric characteristics of the studied ponds, along with the number of traps used and the ecological status values (QAELS and ECELS) for each pond, Table S2: Results of the MLMs (N = 49) showing the predictor variables that affect size-related variables and abundance (CPUE) of *Aphanius iberus*. Both Full models and Best models are presented. For each one, intercept (estimate and standard error, S.E.), Beta coefficients (standardized), *t*-value, significance (*p*-value), and df are shown.

**Author Contributions:** Conceptualization: S.B., A.B. and S.S.; sampling: S.S., A.B., L.B., M.B., I.A. and S.B.; statistical analysis: S.S.; writing—original draft preparation: S.S.; methodology: S.B., S.S., M.B., I.A., L.B., A.B. and writing—review and editing: S.B., S.S., M.B., I.A., L.B., A.B. All authors have read and agreed to the published version of the manuscript.

**Funding:** This research was performed within the framework of the "SOS Fartet Project" supported by Fundación Biodiversidad and is also funded by a grant from the Spanish Ministry of Science, Innovation and Universities (grant no. RTI2018-095363-B-I00). S.B. was partially supported by a grant from the Deutsche Forschungsgemeinschaft (DFG) (grant no. Me 1686/7-1).

**Acknowledgments:** We thank Sergi Romero, director of "Aiguamolls de l'Empordà" Natural Park, and Marc Marí from "El Montgrí, les Illes Medes i el Baix Ter" Natural Park for the facilities and authorizations to develop the project. We are also thankful to the researchers and undergraduate students who helped during sampling: S. Carrasco, P. Ortega, M. Carol, P. Antoni, M. Omer, O. Vigil, and E. Corella. Thanks also to Z. Ersoy for her support with the chlorophyll-a and statistical analyses.

**Conflicts of Interest:** The authors declare no conflict of interest. The funding sponsors had no role in the design of the study; in the collection, analyses or interpretation of data; in the writing of the manuscript or in the decision to publish the results.

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
