# Peer review of "Factors Influencing Abundances and Population Size Structure of the Threatened and Endemic Cyprinodont Aphanius iberus in Mediterranean Brackish Ponds"

_water, doi:10.3390/w12113264_

Round 1

Reviewer 1 Report

It appears to me as a paper not really "polished" and with some details to be adjusted :

All latin names must be in italics (abstract, lines122, 123, 125, ...) and species names without capitals (line 224)...

§0 should be suppressed (before introduction)

Why italics, line 198-207 ?

Quite interesting but not very new to discover that a population is in better conditions when environment is of better quality !

Can You add some explanations of the ways environment directly affect population size and clarify the negative impact of Gambusia ?

Author Response

Reviewer 1

Comments and Suggestions for Authors

It appears to me as a paper not really "polished" and with some details to be adjusted :

All latin names must be in italics (abstract, lines122, 123, 125, ...) and species names without capitals (line 224)...

[Response] Revised as requested.

0 should be suppressed (before introduction)

[Response] Revised as requested.

Why italics, line 198-207 ?

[Response] Revised as requested.

Quite interesting but not very new to discover that a population is in better conditions when environment is of better quality !

[Response] We thank the reviewer for revising our manuscript. Nonetheless, we think that our results are relevant because identifying the key factors that influence the population structure of A. iberus can help developing efficient conservation and management plans.

Can You add some explanations of the ways environment directly affect population size and clarify the negative impact of Gambusia ?

[Response] We apologise but we think that this question is not fully clear. Concerning the negative effect of Gambusia, our results suggested that both CPUE and size metrics of A. iberus were NOT affected by the presence of this allochthone fish. Thus, we think it is better not to speculate on the negative impacts that Gambusia might have on Aphanius since they were not observed in our study. Also, in the Introduction, we have already presented some impacts that Gambusia can have on Aphanius which we did not observe here.

Concerning the ways that environment directly affect population size, we have explained how the variables related to the environment and that were found significant in our study affected population size. Please, see lines: 281-288; 294-299; 301-305.

Reviewer 2 Report

The study reported in the ms. related the size structure of Aphanius iberus (a threatened cyprinodont) to the ecological status of Mediterranean brackish ponds

The study encompasses several interest points, including an endangered fish species, an interesting habitat (brackish ponds) and the establishement of important relations between the size-structure of fish populations and the ecological status of the studied ponds (i.e. some abiotic and biotic factors). As the author said, "Identifying the key factors that influence the population structure of A. iberus is relevant to develop efficient conservation and management plans for this endangered species". The study should be of interest to readers of Water.

Overall, the english of the ms. is good and the authors have used an appropriate sampling design.

I have no major comments. Notwithstandingm I would suggest a little more care with some of the statements/phrases made in the text. For example the authors state that A. iberus size structure is related with the ecological status of ponds, but the relation detected only involved one of the two indices used. Consequently, the authors found a relation between the size-structure of the fish and some indicators of ecological status and not with ecological status per se.    

Do the studied ponds with higher ecological status (as assessed with QAELS, Water Quality of Lentic and Shallow Ecosystems) present higher total nitrogen concentrations?

Could you please clarify that when you say that (lines 102-103 "Finally, we would expect that the abundance of A. iberus would be negatively correlated with zooplankton biomass due to predation") you are refering the fish predation on the zooplankton.

Author Response

Reviewer 2

The study reported in the ms. related the size structure of Aphanius iberus (a threatened cyprinodont) to the ecological status of Mediterranean brackish ponds

The study encompasses several interest points, including an endangered fish species, an interesting habitat (brackish ponds) and the establishement of important relations between the size-structure of fish populations and the ecological status of the studied ponds (i.e. some abiotic and biotic factors). As the author said, "Identifying the key factors that influence the population structure of A. iberus is relevant to develop efficient conservation and management plans for this endangered species". The study should be of interest to readers of Water.

Overall, the english of the ms. is good and the authors have used an appropriate sampling design.

[Response] We thank the reviewer for the positive feedback of our study.

I have no major comments. Notwithstandingm I would suggest a little more care with some of the statements/phrases made in the text. For example the authors state that A. iberus size structure is related with the ecological status of ponds, but the relation detected only involved one of the two indices used. Consequently, the authors found a relation between the size-structure of the fish and some indicators of ecological status and not with ecological status per se.

[Response] We agree with the reviewer. In fact, the ECELS index is more related to the conservation status of the ponds because it is based on morphological aspects, type of aquatic vegetation and human impacts, whereas QAELS is related to the water quality and is based on the composition of microcrustacean assemblages and taxonomic richness of aquatic insects and crustaceans. We have clarified this, and we also specified in the text that the relationship was found with QAELS index.

Do the studied ponds with higher ecological status (as assessed with QAELS, Water Quality of Lentic and Shallow Ecosystems) present higher total nitrogen concentrations?

[Response] We computed the correlation of the QAELS index and total nitrogen, and they were weekly correlated (Pearson correlation value: 0.08, p-value: 0.79).

Could you please clarify that when you say that (lines 102-103 "Finally, we would expect that the abundance of A. iberus would be negatively correlated with zooplankton biomass due to predation") you are refering the fish predation on the zooplankton.

[Response] Revised as requested. You can find the change in the new version of the manuscript at lines 97-98.

Reviewer 3 Report

Dear Authors,

The paper is well written in large parts. In my eyes, it is important and interesting to understand better the needs and habitat use of threatened species, otherwise it would not be possible to protect them or to make possible their survival. Your study gives insight into interactions between biological traits of the Spanish toothcarp and environmental parameters of their still existing habitats.

I found only a few shortcomings (see remarks in the pdf document). But in order to improve the manuscript, I would strongly suggest the following changes:

Line 2-4: It is necessary to reconsider the title of this paper. The current title focuses on one finding of the study only. Therefore, it and is rather misleading. Moreover, it comes near to the results of the paper by Casas et a. (2011) in Biological Conservation.

A title like "Factors influencing abundances and population size structure of A. iberus" would fit much better the current paper and focuses more on the approach of your study that is quite different from that by Casas et al.

Line 19 and further parts in the abstract and discussion: The term "confinement" is an unfortunate choice in your manuscript. What you mean is the isolation of ponds.

Furthermore, I suggest to include two photographs of typical habitats of this fish that gives an impression to those readers who do not know the habitats of this endemic species. I hope that my comments will contribute to improve your paper,

sincerely yours,

a reviewer

Author Response

Reviewer 3

The paper is well written in large parts. In my eyes, it is important and interesting to understand better the needs and habitat use of threatened species, otherwise it would not be possible to protect them or to make possible their survival. Your study gives insight into interactions between biological traits of the Spanish toothcarp and environmental parameters of their still existing habitats.

I found only a few shortcomings (see remarks in the pdf document).

[Response] We thank the reviewer for the interest in our manuscript.

But in order to improve the manuscript, I would strongly suggest the following changes:

Line 2-4: It is necessary to reconsider the title of this paper. The current title focuses on one finding of the study only. Therefore, it and is rather misleading. Moreover, it comes near to the results of the paper by Casas et a. (2011) in Biological Conservation.

A title like "Factors influencing abundances and population size structure of A. iberus" would fit much better the current paper and focuses more on the approach of your study that is quite different from that by Casas et al.

[Response] We thank the reviewer for the new proposal of the title.  Accordingly, we changed the title in order to focus on a broader perspective of our study.

Line 19 and further parts in the abstract and discussion: The term "confinement" is an unfortunate choice in your manuscript. What you mean is the isolation of ponds.

[Response] We agree with the reviewer. In the revised version, we replaced the term ‘confinement’ with ‘isolation’ throughout the manuscript.

Furthermore, I suggest to include two photographs of typical habitats of this fish that gives an impression to those readers who do not know the habitats of this endemic species.

[Response] Thank you for the good advice. We provided two pictures from the studied ponds (Figure S2 in Supplementary materials) and the corresponding citation in the main text (line 104-105).
